# A multiplexed DNA FISH strategy for assessing genome architecture in *Caenorhabditis elegans*

Brandon D Fields[1,2†], Son C Nguyen[2†‡], Guy Nir[2], Scott Kennedy[2]*

[1]Laboratory of Genetics, University of Wisconsin-Madison, Madison, United States; [2]Department of Genetics, Harvard Medical School, Boston, United States

**Abstract** Eukaryotic DNA is highly organized within nuclei and this organization is important for genome function. Fluorescent *in situ* hybridization (FISH) approaches allow 3D architectures of genomes to be visualized. Scalable FISH technologies, which can be applied to whole animals, are needed to help unravel how genomic architecture regulates, or is regulated by, gene expression during development, growth, reproduction, and aging. Here, we describe a multiplexed DNA FISH Oligopaint library that targets the entire *Caenorhabditis elegans* genome at chromosome, three megabase, and 500 kb scales. We describe a hybridization strategy that provides flexibility to DNA FISH experiments by coupling a single primary probe synthesis reaction to dye conjugated detection oligos via bridge oligos, eliminating the time and cost typically associated with labeling probe sets for individual experiments. The approach allows visualization of genome organization at varying scales in all/most cells across all stages of development in an intact animal model system.
DOI: https://doi.org/10.7554/eLife.42823.001

*For correspondence:
kennedy@genetics.med.harvard.edu

†These authors contributed equally to this work

Present address: ‡Department of Genetics, Penn Epigenetics Institute, Perelman School of Medicine, University of Pennsylvania, Philadelphia, United States

Competing interests: The authors declare that no competing interests exist.

## Introduction

Eukaryotic genomes are non-randomly organized within mitotic and interphase nuclei. The basic unit of genome organization is the nucleosome. Nucleosomes assemble into higher order structures whose biogenesis, maintenance, regulation, and purpose are poorly understood (*Bickmore, 2013*; *Bonev and Cavalli, 2016*; *Meaburn, 2016*; *Yu and Ren, 2017*). DNA fluorescent *in situ* hybridization (FISH) technologies and chromatin conformation capture techniques allow 3D architectures of genomes to be assessed (*Bauman et al., 1980*; *Beliveau et al., 2012*; *Bienko et al., 2013*; *Dekker et al., 2013*, *Dekker et al., 2002*; *Lieberman-Aiden et al., 2009*). Studies using these technologies have begun to reveal how DNA is organized within nuclei. For instance, chromatin capture experiments show that many eukaryotic genomes are assembled into megabase-sized structures termed topologically associated domains (TADs). TADs, and larger organizational units termed compartments, are thought to allow subregions of chromosomes to share and integrate long-range transcriptional regulatory signals (*Dekker and Heard, 2015*; *Dekker and Mirny, 2016*; *Dixon et al., 2012*; *Lieberman-Aiden et al., 2009*; *Vernimmen and Bickmore, 2015*). Additionally, DNA FISH and chromatin immunoprecipitation (ChIP) experiments have shown that the position of genes within nuclei is often not random: active genes tend to localize near nuclear pores and/or the nuclear interior while inactive genes tend to localize to the nuclear periphery, distant from nuclear pores (*Casolari et al., 2004*; *Gonzalez-Sandoval and Gasser, 2016*; *Lemaître and Bickmore, 2015*; *Pickersgill et al., 2006*; *van Steensel and Belmont, 2017*). Finally, DNA FISH experiments have shown that individual chromosomes tend to occupy distinct non-overlapping regions of subnuclear space, even in interphase nuclei (termed chromosome territories) (*Bolzer et al., 2005*; *Cremer and Cremer, 2010*). Many questions concerning the large-scale architecture of genomes remain unanswered, including the following: how the various aspects of genome architecture, such as gene

**eLife digest** DNA contains the instructions needed to build and maintain a living organism. How DNA is physically arranged inside a cell is not random, and DNA organization is important because it can affect, for example, which genes are active, and which are not.

Researchers often use a technique called "fluorescence *in situ* hybridization" (or FISH for short) to study how DNA is organized in cells. FISH tethers fluorescent molecules to defined sections of DNA, making those sections glow under the right wavelength of light. It is possible to collect images of the fluorescent DNA regions under a microscope to see where they are in relation to each other and to the rest of the cell.

Fields, Nguyen et al. have now created a new library of FISH molecules that can be used to analyze the DNA of a microscopic worm known as *Caenorhabditis elegans* – a model organism that is widely used to study genetics, animal development, and cell biology. The library can be used to visualize the worm's whole genome at different scales. The library enables accurate and reliable investigations of how DNA is organized inside *C. elegans*, including in intact worms, meaning it also offers the first chance to study DNA organization in a whole organism through all stages of its life cycle.

This new resource could help to reveal the relationships between DNA organization, cell specialization and gene activity in different cells at different stages of development. This could help to clarify the relationships between physical DNA organization and biological change. This design strategy behind this whole genome library should also be adaptable for similar studies in other animal species.

DOI: https://doi.org/10.7554/eLife.42823.002

position, TADs, or territories differ in different cell types or across developmental time, and how such changes relate to gene expression during development. Technologies that enable rapid and flexible analysis of genome organization in an intact animal would allow such questions to be addressed.

The nematode *Caenorhabditis elegans* is an excellent model for studying genome organization in an intact animal due to its size (1 mm), lifespan (~3 days to reproductive maturity), genome size (100 Mb across five autosomes and one sex chromosome), and transparent body. Whole-chromosome DNA FISH experiments have been instrumental for our current understanding of chromosome architecture and dynamics in model organisms such as *C. elegans* (*Lau et al., 2014*; *Nabeshima et al., 2011*). Such studies are limited, however, by the cost and time associated with generating DNA FISH probe sets. For example, a previous chromosome level DNA FISH experiment in *C. elegans* used 127 yeast artificial chromosome (YAC) clones split into 50 amplification reactions for three chromosome probe generation (*Nabeshima et al., 2011*). Oligopaint technology has made DNA FISH probe production faster, cheaper, and more flexible (*Beliveau et al., 2012*). Oligopaint takes advantage of massively parallel DNA synthesis technologies to create user defined libraries containing hundreds of thousands of individual DNA oligos each comprised of a short (42 bp) DNA sequence that hybridizes to a genome, as well as additional 'barcode' sequences that serve two major functions. First, barcodes allow probes to be repeatedly amplified from an Oligopaint library, thus providing a virtually inexhaustible supply of oligos for DNA FISH (*Beliveau et al., 2012*; *Chen et al., 2015*; *Murgha et al., 2014*). Second, barcodes allow pre-labeled detection oligos to be used to detect Oligopaint oligos, thus obviating the need for fluorescently labeling probes for each DNA FISH experiment (*Beliveau et al., 2015*; *Beliveau et al., 2012*; *Nir et al., 2018*; *Rosin et al., 2018*).

Previously, the Oligopaint technology has been used to visualize chromosome territories in *Drosophila* cultured cells in normal and mutant contexts (*Rosin et al., 2018*). Here, we describe a rapid, flexible and inexpensive Oligopaint strategy that enables visualization of chromosome territories and sub-chromosome regions in a whole intact organism. Specifically, we report methods for using this library to simultaneously visualize all six *C. elegans* chromosomes, as well as three megabase and 500 kilobase subregions of these chromosomes, in all/most cells of *C. elegans* across all stages of development.

# Results

## *C. elegans* oligopaint library design.

An Oligopaints bioinformatics pipeline was used to identify 42 bp DNA sequences in the *C. elegans* genome (genome reference Ce10) that (1) uniquely map to the genome, (2) exhibit similar melting temperatures and similar GC content, (3) lack repetitive stretches, and (4) lack predicted secondary structures (*Beliveau et al., 2012*). [Note, an updated bioinformatics pipeline for identifying probes is now available (*Beliveau et al., 2018*).] The results of our bioinformatic search revealed approximately nine suitable probe sequences per kilobase of *C. elegans* genomic DNA (*Table 1*). We generated an Oligopaint library that contained, on average,~2 probe sequences per kb of genomic DNA across each *C. elegans* chromosome (*Figure 1a* and *Table 1*). *Figure 1a* shows the distribution of all (25,174) chromosome I probes. Few gaps between probes exceeded 5 kb (33 out of 25,174), with the largest gap spanning ~18 kb. A subtle decrease in probe density is observed on chromosome arms, perhaps due to an increase in repetitive sequences in these regions, which biased against probe selection (C. elegans *C. elegans Sequencing Consortium, 1998*). Similar probe distributions are observed for the other five *C. elegans* chromosomes (*Figure 1—figure supplement 1*). In total, the library consisted of 170,594 oligos (termed primary Oligopaint oligos), which each contain 42 bp of unique genomic sequence flanked by barcode sequences that allow for DNA FISH targeting each of the six *C. elegans* chromosomes, as well as three megabase, or 500 kb subregions of these chromosomes (*Figure 1b*). Bridge oligos (also see *Nir et al., 2018*) were designed to base pair with barcode sequences contained within primary probes as well as base pair with dye-conjugated detection oligos (*Figure 1c*). Detection oligos were designed that base pair with bridge oligos and are conjugated to three fluorophores (Alexa 488, Cy3, and Alexa 647) (*Figure 1c*). Thus, bridge oligos are intermediate probes that hybridize to the primary probe and provide a docking site for labeled detection probes. Bridge oligos provide versatility (and cost savings) to DNA FISH experiments as these oligos allow any primary probe set to be coupled to any detection probe set with minimal additional cost. Bridge oligos also allow for more than one fluorophore to be targeted to primary probes, which expands the number of objects that can be visualized with a standard three channel microscopy system (see six chromosome FISH experiments below). By using 1) detection oligos with fluorophores on both 3' and 5' termini, 2) two detection oligos per bridge oligo, and 3) bridge oligos that target the 5' and 3' barcode sequence of primary probes, it is possible to have each primary oligo recognized by eight fluorophores. To conduct DNA FISH, unlabeled primary probes are first PCR amplified from the Oligopaint library as described in *Beliveau et al. (2017)*; *Beliveau et al., 2012*; *Chen et al., 2015*; *Murgha et al., 2014* (also see Materials and methods). Second, primary Oligopaint oligos are hybridized to fixed samples of *C. elegans* overnight (see below and Materials and methods). Third, samples are hybridized with a mixture of bridge oligos and dye-conjugated

**Table 1.** Distribution of oligonucleotide sequences across the *C. elegans* genome.
Potential probe sequences were identified as described in Materials and methods. Every fifth oligo sequence was incorporated into the library to ensure even distribution. The average distance between selected probes for each chromosome, and the standard deviations of these distances, are indicated. Kb, kilobase. bp, base pair.

| Chr | First coordinate | Last coordinate | Potential oligo sequences | Potential oligo sequences/ kb | Oligo sequences chosen | Oligo sequences chosen/kb | Average distance between oligos (bp) | Standard deviation of distance between oligos (bp) |
|---|---|---|---|---|---|---|---|---|
| I | 512 | 14999984 | 125,863 | 8.39 | 25,174 | 1.68 | 595.8 | 505.9 |
| II | 163 | 15199262 | 136,793 | 9.00 | 27,360 | 1.80 | 555.5 | 484.5 |
| III | 123 | 13599784 | 133,977 | 9.85 | 22,796 | 1.68 | 595.9 | 572.2 |
| IV | 655 | 17399857 | 137,270 | 7.89 | 27,454 | 1.58 | 633.8 | 585.4 |
| V | 414 | 20699907 | 177,948 | 8.60 | 35,590 | 1.72 | 581.6 | 460.4 |
| X | 595 | 17599761 | 161,095 | 9.15 | 32,220 | 1.83 | 546.2 | 436.8 |
| | | | 872,946 | 8.81 | 170,594 | 1.71 | 584.8 | 507.5 |

DOI: https://doi.org/10.7554/eLife.42823.003

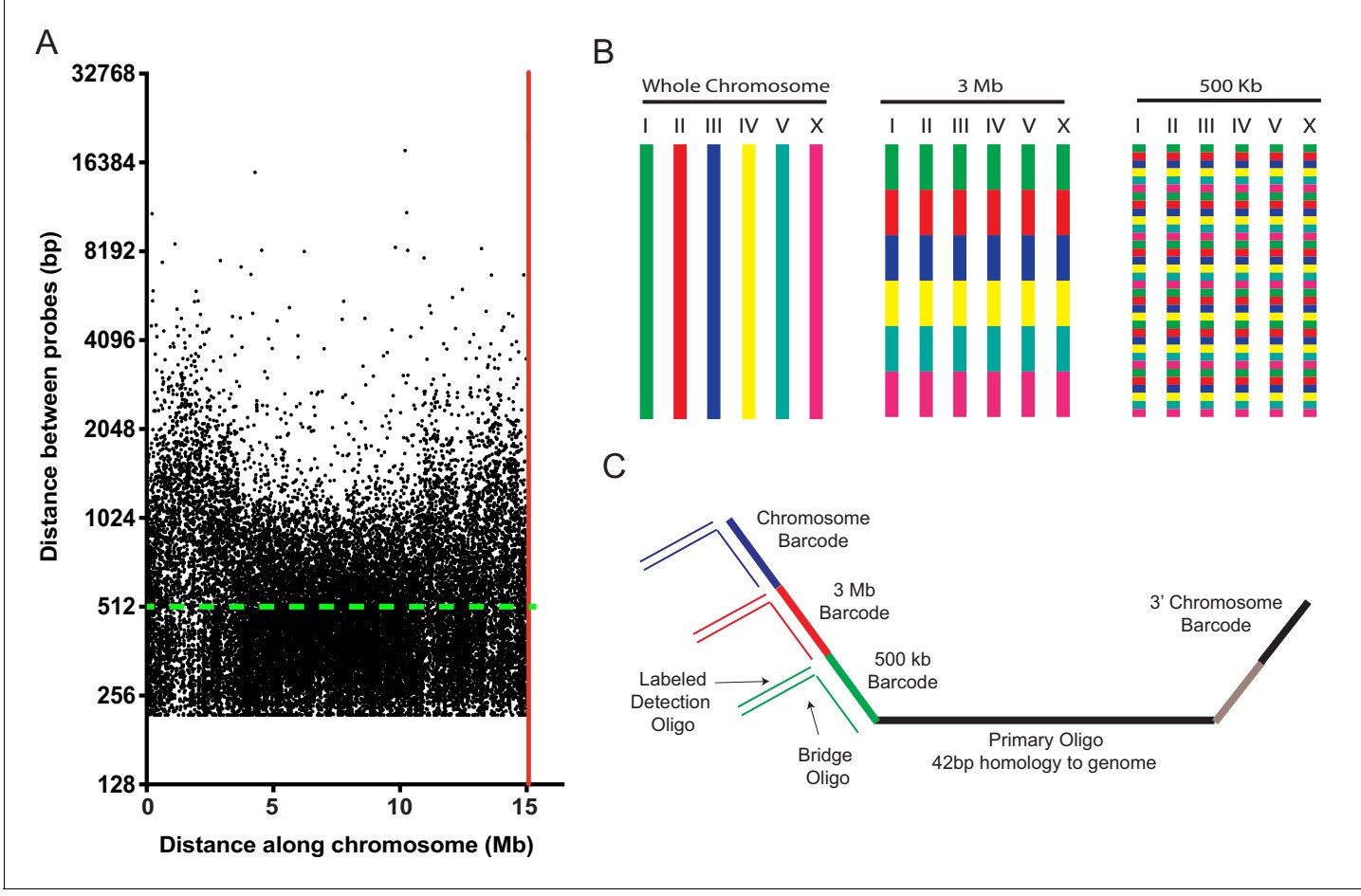

**Figure 1.** A highly multiplexed oligo library for *C. elegans* Oligopaint. (**A**) The distribution of Oligopaint probes across chromosome I is shown. Green dashed line indicates the two probe/kb average. Red line indicates the end of the chromosome. Probe distribution is similar for other chromosomes (see *Table 1* and *Figure 1—figure supplement 1*). (**B**) Oligopaint library allows primary oligos, which are specific to any chromosome, 3 Mb, or 500 Kb region within any chromosome, to be specifically amplified. Primary probes are PCR amplified from Oligopaint library and produced as described in Materials and methods. Bridge and detection probes allow the indicated chromosomal regions to be visualized. (**C**) Primary probes consist of barcode sequences appended to 42 bp sequences that hybridize uniquely to the *C. elegans* genome. Total length of each oligo is 150 bp. Barcode sequences allow each primary probe to be amplified as part of a pool of primary probes that target a chromosome (chromosome barcode), 3 Mb subsection of chromosome (3 Mb barcode), or 500 kb subsection of chromosome (500 kb barcode). Bridge oligos and Detection oligos (arrows) are used to recognize and illuminate primary probes. Note: primary oligos contain an additional barcode not used in this work (brown). The barcode is specific to each 500 kb segment and could be used to increase detection efficiencies of 500 kb DNA FISH by allowing an addition detection oligo to be incorporated during the detection phase of DNA FISH.

DOI: https://doi.org/10.7554/eLife.42823.004

The following figure supplement is available for figure 1:

**Figure supplement 1.** Oligopaint probe distribution along all six *C. elegans* chromosomes.

DOI: https://doi.org/10.7554/eLife.42823.005

detection oligos for 3 hr the following day (*Figure 1c*). Together, this three-step strategy allows many DNA FISH experiments to be conducted fairly cheaply after a single primary probe synthesis step.

### *C. elegans* oligopaint staining is robust and specific

DNA FISH in *C. elegans* is typically done on dissected tissue. We developed a fixation and hybridization protocol that allowed for efficient DNA FISH on intact *C. elegans*. As part of this protocol, hybridization steps are conducted in microcentrifuge tubes, which allows large numbers of animals to be simultaneous assayed by FISH. A detailed description of this fixation and hybridization

protocol can be found in Materials and methods. To test our *C. elegans* Oligopaint library, we amplified a primary probe set targeting chromosome II (27,360 unique probes) and asked if this probe set was able to specifically label chromosome II. The behavior and morphology of chromosomes in the *C. elegans* germline are well-established (*Albertson et al., 2011*). For instance, homologous chromosomes pair at the pachytene stage of Meiosis I at a defined region of the germline (termed pachytene region). Oocytes are arrested in diakinesis of meiosis I and chromosomes are highly condensed with homologs connected via a single chiasmata (termed bivalents) (*Villeneuve, 1994*). Mature sperm harbor highly condensed chromosomes and are haploid. To address specificity, we imaged germ cells of animals subjected to chromosome II DNA FISH. This analysis detected the expected chromosomal structures in pachytene germ cells, oocytes, and sperm; fluorescent staining was observed on a single bivalent in oocytes and in one region of the nucleus in sperm and

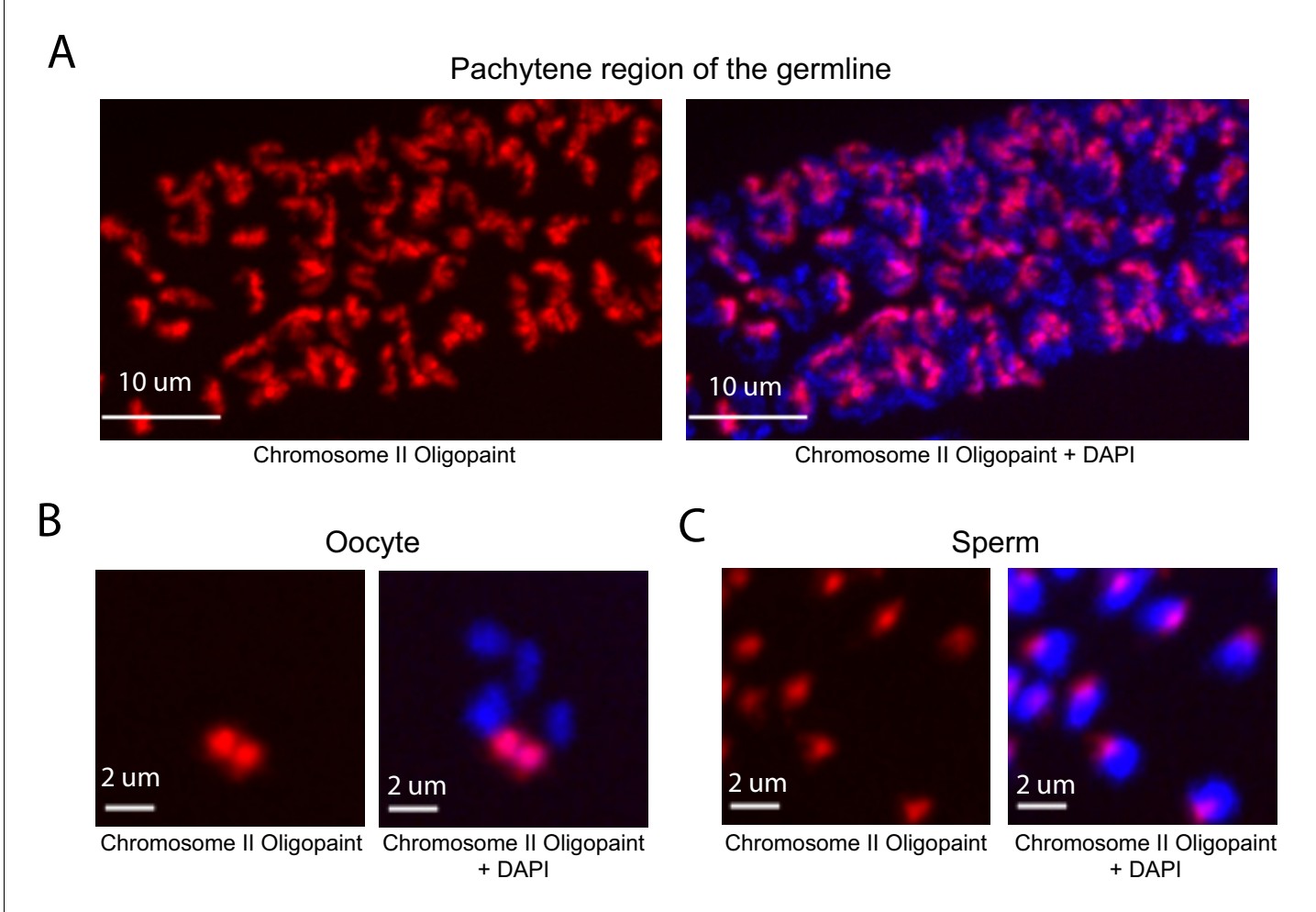

**Figure 2.** Whole chromosome Oligopaint in *C. elegans* is specific. (A–C) Adult *C. elegans* were fixed as described in Materials and methods. Fixed animals were subjected to three step hybridization to detect Chromosome 2 (red). Animals were co-stained with DAPI (blue). (**A**) 3D maximum projection of the pachytene region of an adult *C. elegans* germline is shown. (**B**) 3D maximum projection of an oocyte. A single bivalent is stained with chromosome II Oligopaints. (**C**) 3D maximum projection of sperm. Scale bars for images are indicated. All images are representative of at least three independent animals.

DOI: https://doi.org/10.7554/eLife.42823.006

The following figure supplements are available for figure 2:

**Figure supplement 1.** X chromosome Oligopaint on *him-8 (e1489)* mutant animals.
DOI: https://doi.org/10.7554/eLife.42823.007

**Figure supplement 2.** Oligopaints stain chromosomes in all/most somatic cells and in all/most developmental stages of *C. elegans*.
DOI: https://doi.org/10.7554/eLife.42823.008

pachytene germ cells (*Figure 2*). HIM-8 is required for X chromosome homolog pairing during meiosis and, consequently, X chromosomes are present as two univalents (and not a single bivalent) in *him-8* mutant animals (*Phillips et al., 2005*). To further address specificity, we imaged oocytes of wild type and *him-8 (e1489)* mutant animals subjected to X chromosome Oligopaint. As expected, X chromosome DNA FISH stained a single bivalent in wild-type oocytes (58/58) and two univalents in 84% of *him-8* oocytes (51/61) (*Figure 2—figure supplement 1*). Together the data show that the *C. elegans* Oligopaint library is specific.

To quantify the efficiency of our method, we first measured the percentage of whole animals stained by chromosome II DNA FISH. Staining presented in an all or nothing fashion with 1085/1303 (83%) of larval stage animals, 317/326 (97%) of adult animals, and 50% of the embryos housed within uteri of adult animals displaying FISH signal (*Figure 2—figure supplement 2*). [Note, an alternative protocol that allows for greater efficiency using isolated embryos (90%) is described in Materials and methods.] We next measured the % of nuclei within a given animal that were stained by chromosome II DNA FISH. We randomly chose DAPI-stained nuclei (from animals that showed staining) and asked if these nuclei were positive for chromosome II DNA FISH signals. Out of 50 randomly chosen somatic nuclei 50/50 had DNA FISH signal. Likewise, 50/50 germline nuclei were positive for DNA FISH signals. We conclude that our library and hybridization strategy allows for robust and specific labeling of a whole chromosome in many cell types and many developmental stages simultaneously in large numbers of animals. It is possible that DNA FISH signals in every cell and at every stage of development can be visualized with this approach.

## Simultaneous visualization of all six *C. elegans* chromosomes

*C. elegans* possess six chromosomes. Most lab microscopy systems are equipped to image 3-4 fluorophores and, thus, are not capable of imaging all six *C. elegans* chromosomes simultaneously. To circumvent this issue, we took advantage of our bridge oligo strategy to target combinations of fluorophores to each *C. elegans* chromosome in order to image all six chromosomes using a three-channel microscopy system (*Figure 3*). For instance, *Figure 3—figure supplement 1* shows an example of multi-probe labeling of the X chromosome: Detection oligos labeled with either Alexa 647 or Cy3 were targeted to the X chromosome (*Figure 3—figure supplement 1a*). Overlapping Alexa 647 (green) or Cy3 (red) channels produces a yellow pseudocolor that can be differentiated from Alexa 647 (green) or Cy3 (red) alone (*Figure 3—figure supplement 1b*). We next amplified primary probe sets targeting all six *C. elegans* chromosomes and hybridized primary probe sets to fixed adult *C. elegans*. We then used bridge oligos to couple these primary probe sets to detection probes labeled with Alexa 488, Cy3, or Alexa 647, or combinations of these fluorophores, in order to simultaneously image all six *C. elegans* chromosomes (*Figure 3a*). We imaged oocytes in these animals and detected six bivalents that were each labeled a distinct color (*Figure 3b*). In pachytene stage adult germ cells, *C. elegans* chromosomes are paired, condensed, and localized near the nuclear periphery (*Albertson et al., 2011*). DNA FISH illuminated six regions of distinct colors concentrated near the nuclear periphery in pachytene germ cells (*Figure 3c*). Six chromosome DNA FISH staining was robust: 50/50 randomly chosen DAPI positive nuclei were stained successfully. Six chromosome FISH staining was also successful in somatic nuclei. Six distinct colors were often distinguishable in the nuclei of intestinal and hypodermal nuclei, as well as nuclei whose small size and positioning within the animal were indicative of ventral cord neurons (*Figure 3d–f* and *Figure 3—video 1*). These data show that our DNA FISH approach is capable of labeling all six *C. elegans* chromosomes simultaneously in many different cell types of an intact animal. The data also show that, like interphase chromosomes in other eukaryotes, *C. elegans* chromosomes occupy largely distinct territories within interphase nuclei and that these chromosome territories persist in post-mitotic cells. Note, because the six color strategy described above visualizes an overlap of two colors (each of which is also used to define a separate chromosome), rigorously defining the subnuclear space occupied by individual chromosomes is not possible using this six color approach. Single channel probe sets are recommended for experiments in which knowing the precise space occupied by a chromosome is relevant (see Figure 5 for examples).

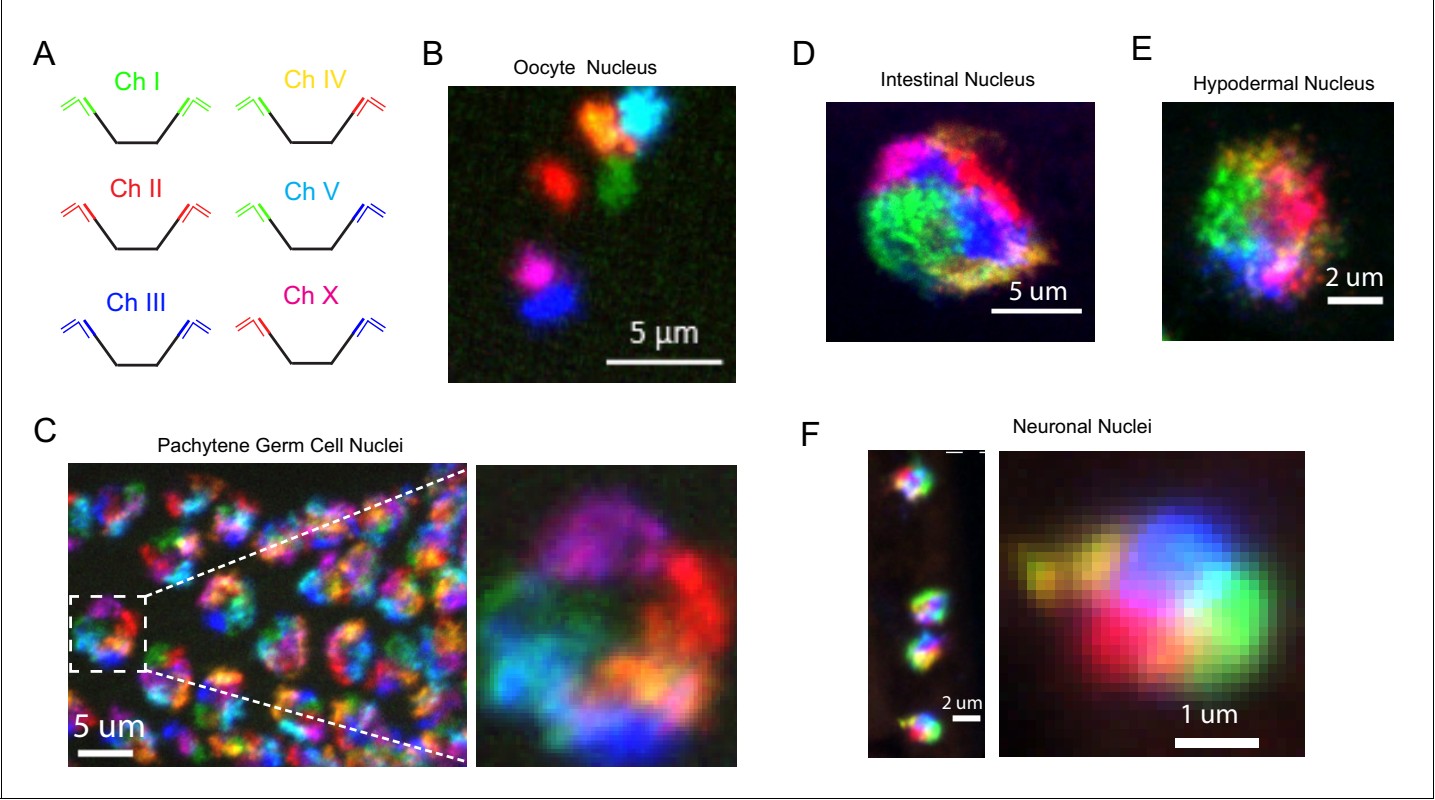

**Figure 3.** Simultaneous visualization of all six *C. elegans* chromosomes. (A) Strategy to detect six chromosomes is shown. Detection probes labeled with Alexa488 (Green), Cy3 (Red), and Alexa647 (Blue), or combinations of these three fluorophores were used to label each of the six *C. elegans* chromosomes a different color. (B–F) Adult *C. elegans* were fixed and subjected to three step hybridization to detect Chromosomes 1, 2, 3, 4, 5, and X. (B) 3D maximum projection of an oocyte. Each bivalent is labeled a different color. (C) 3D maximum projection of the pachytene region of an adult germline. A magnification of one of these nuclei is shown to the right. (D) 3D maximum projections of an intestinal nucleus, (E) hypodermal nucleus, and (F) nuclei whose size and position within the animal suggest the cell is a ventral cord neuron. All images are representative of at least three independent animals.

DOI: https://doi.org/10.7554/eLife.42823.009

The following video and figure supplement are available for figure 3:

**Figure supplement 1.** Multicolor labeling of the X chromosome to expand the number of detectable objects.
DOI: https://doi.org/10.7554/eLife.42823.010
**Figure 3—video 1.** Six Chromosome FISH of a *C. elegans* intestinal nucleus.
DOI: https://doi.org/10.7554/eLife.42823.011

## Detection of 3 Mb and 500 kb chromosomal subregions

We designed our Oligopaint library to include 3 Mb and 500 Kb barcode sequences that should permit visualization of chromosomal subregions (*Figure 1c*). To test this aspect of our library, we amplified Oligopaint oligos targeting chromosome I and hybridized these probes to adult *C. elegans*. We then used bridge oligos that recognized all Chromosome I Oligopaint oligos (~15 Mb), a 3 Mb subregion of chromosome I (0–3 Mb), or a 500 kb subregion of this 3 Mb region (1.0–1.5 Mb). Detection oligos coupled to Alexa 488, Cy3, and Alexa 647 were used to illuminate each genomic region, respectively. We imaged pachytene germ cells and, as expected, observed a single contiguous DNA FISH signal after chromosome I DNA FISH (*Figure 4a*). 3 Mb DNA FISH illuminated a subregion of the chromosome I signal and 500 kb DNA FISH illuminated a subregion of this 3 Mb signal (*Figure 4a*). Staining was robust, with 50/50 randomly chosen nuclei possessing all three fluorescent signals in successfully stained animals. Similar patterns were observed when chromosome IV, a 3 Mb subregion of chromosome IV (0–3 Mb), or a 500 kb subregion of this 3 Mb region (2.5–3.0 Mb) were analyzed (*Figure 4b*). We conclude that the Oligopaint library has the capability to visualize 3 Mb and 500 kb subregions of the *C. elegans* genome.

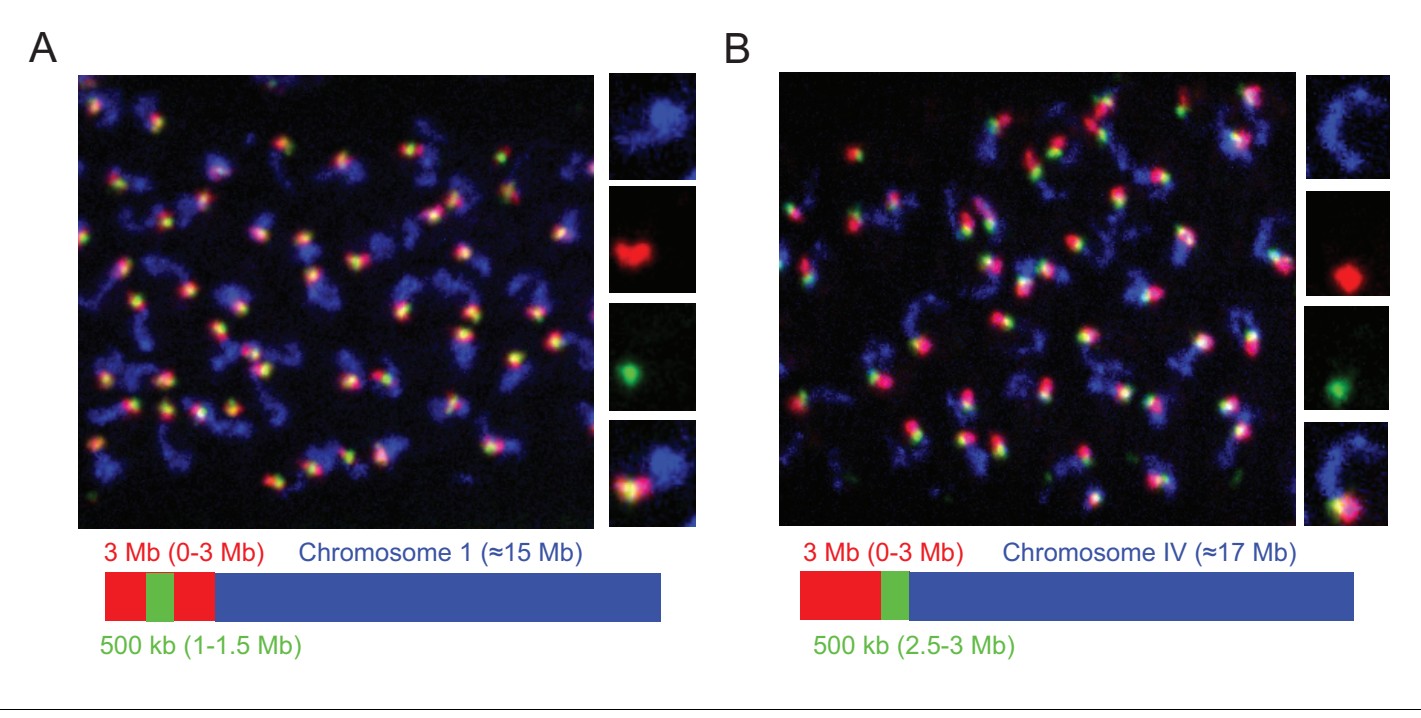

**Figure 4.** Detection of 3 Mb and 500 kb chromosomal subregions. Adult *C. elegans* were fixed and subjected to three step hybridization to detect Chromosome I (**A**) or IV (**B**). (**A–B**) Top, 3D maximum project of pachytene region of adult germline. (**A**) Chromosome I or (**B**) chromosome IV, a 3 MB (0–3 Mb) region of these chromosomes, and a 500 kb (1–1.5 Mb for Ch I and 2.5–3 Mb for Ch IV) region of these chromosomes were targeted with detection probes shown in blue, red, and green, respectively. Magnifications of representative nuclei are shown to the right. Bottom, graphic representations of regions of chromosome I (**A**) or chromosome IV (**B**) that were stained in the experiment are shown. All images are representative of at least three independent animals.

DOI: https://doi.org/10.7554/eLife.42823.012

## Using *C. elegans* oligopaints to explore the biology of genome architecture

Our *C. elegans* Oligopaint library and hybridization protocol should allow many questions relating to the biology of genome organization to be asked within the context of a whole animal. We started this process by using our library to ask two simple questions: 1) Whether genomic architecture changes during aging, and (2) what cellular factors are needed to establish and/or maintain chromosome territories in post-mitotic cells?

Recent studies suggest that higher order chromatin structures may break down during aging, and in aging-related diseases such as Alzheimer's (*Winick-Ng and Rylett, 2018*). Age-related alterations in nuclear morphology have also been noted in *C. elegans* (*Haithcock et al., 2005*). We used our Oligopaint library to simultaneously visualize all six chromosomes in 1- and 10-day-old animals (*C. elegans* typically live about 2 weeks) to ask if the aging process might affect the genomic architecture. For this analysis, we imaged intestinal nuclei as these cells are postmitotic, have large nuclei, and are easily identifiable due to their idiosyncratic size, shape, and location within the animal. As expected, all six chromosomes occupied largely distinct territories in intestinal cells of 1-day-old animals (*Figure 5a*). Interestingly, in 10-day-old animals, chromosomes were no longer organized into discrete territories (*Figure 5a*). Quantification (see Materials and methods) of the space occupied by chromosomes I, II and III in young and aged animals revealed an ~50% increase in the volume of all three chromosomes (*Figure 5b*) as well as an increase in the degree to which chromosomes I, II and III overlapped in subnuclear space (*Figure 5—figure supplement 1a*). The data confirm that the discrete chromosome territories observed in young intestinal nuclei are lost as the worms ages. We asked if the loss of chromosome territories in older animals were a consequence of aging or a function of time. To do so, we conducted a similar analysis on young and old animals harboring a

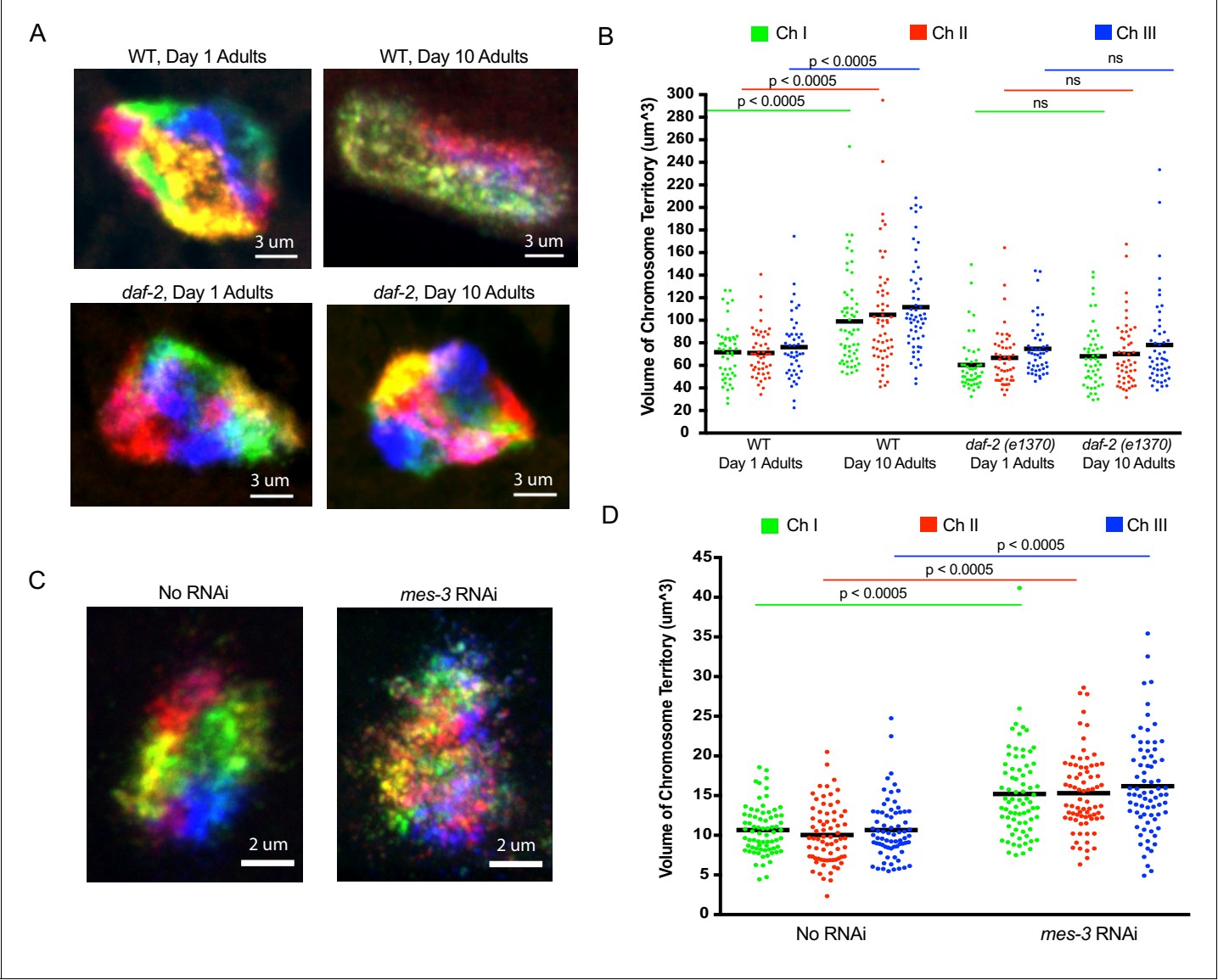

**Figure 5.** Using *C. elegans* Oligopaints to explore the biology of genome organization in a whole animal. (**A**) Adult *C. elegans* were fixed and subjected to three step hybridization to detect all six chromosomes at day 1 or day 10 of adulthood. 3D maximum projections of a representative intestinal nucleus is shown. Territories appear disorganized in ten-day-old animals. Chromosome territories in ten-day-old *daf-2(e1370)* (strain = CB1370) animals are not disorganized. Images shown are representative of at least three independent animals across three biological replicates. (**B**) Experiment was repeated to detect just chromosomes I, II, and III (this was done to allow unambiguous identification of all three chromosomes as six color DNA FISH introduces overlapping fluorescence signals). ImageJ was used to quantify chromosome territory volume by chromosomes I, II, and III in 1- and 10-day-old animals (see Materials and methods for details of quantification). The volume of chromosomes I, II, and III in 1- or 10-day-old animals of the indicated genotype is indicated on the y-axis. Black bar = mean value of all data points. p-Values were calculated using a two-tailed student t-test. n =>50 nuclei from at least five animals. (**C**). Adult *C. elegans* were subjected to *mes-3* RNAi by feeding animals bacteria expressing *mes-3* dsRNA for two generations. Animals were subjected to three step hybridization to detect all six chromosomes and 3D maximum projections of a hypodermal nucleus is shown. Territories appear disorganized after treatment with *mes-3* dsRNA. Images shown are representative of at least three independent animals across two biological replicates. (**D**) Quantifications of two independent three chromosome DNA FISH experiments reveals an increase in chromosome territory volume after *mes-3* dsRNA. Black bar = mean value of all data points. p-Values were calculated using a two-tailed student t-test. n =>50 nuclei from at least five animals.

DOI: https://doi.org/10.7554/eLife.42823.013

The following source data and figure supplements are available for figure 5:

**Source data 1.** RNAi clones tested for effects on chromosome territory architecture.

DOI: https://doi.org/10.7554/eLife.42823.016

**Source data 2.** Raw data for *Figure 5*.

*Figure 5 continued on next page*

*Figure 5 continued*

DOI: https://doi.org/10.7554/eLife.42823.017

**Source data 3.** Raw data for *Figure 5—figure supplement 1*.

DOI: https://doi.org/10.7554/eLife.42823.018

**Figure supplement 1.** Aged animals and *mes-3* RNAi-treated animals display increased overlap in chromosome territories.

DOI: https://doi.org/10.7554/eLife.42823.014

**Figure supplement 2.** Additional cell types for six chromosome FISH on *mes-3* RNAi.

DOI: https://doi.org/10.7554/eLife.42823.015

mutation (*e1370*) in *daf-2. daf-2* encodes a insulin-like receptor and loss-of-function mutations in *daf-2* and mutations in *daf-2* such as *e1370* cause animals to live twice as long as wild-type animals (*Kenyon et al., 1993*; *Kimura et al., 1997*). Chromosome territories did not become enlarged or disorganized in 10-day-old *daf-2* mutant animals versus young animals, indicating that the loss of chromosome territories we see in older wild-type animals is linked to aging and not chronological time (*Figure 5a/b* and *Figure 5—figure supplement 1a*). The data are consistent with a model in which higher order chromatin structures are lost during aging. Further studies will be needed to address if genome organization in other/all cell types is similarly affected by aging and, more importantly, if the loss of chromosomal territories that occur in aged animals is a cause or consequence of the aging process.

Very little is known about how chromosome territories are established or maintained in animals. The Oligopaint DNA FISH library described above could be used to identify and characterize genes and pathways mediating and regulating these processes. As a first attempt to identify such factors, we conducted six chromosome DNA FISH on animals subjected to RNAi targeting seven candidate genes, which we suspected might be involved in establishing/maintaining chromosome territories in *C. elegans* (*Figure 5—source data 1*). *mes-3*, which encodes a component of the Polycomb Repressive Complex 2 (PRC2), was included in this small-scale screen because PRC2 is a known regulator of chromatin architecture in many organisms (*Capowski et al., 1991*; *Holdeman et al., 1998*; *Ross and Zarkower, 2003*; *Xu et al., 2001*). RNAi targeting *mes-3* caused a loss of chromosome territories in adult hypodermal cells (*Figure 5c*). Note: we chose to image hypodermal cells for this analysis as these cells are, like intestinal cells, easy to identify and because (for unknown reasons) the effects of *mes-3* RNAi on genome architecture appeared to be most dramatic in this cell type (*Figure 5—figure supplement 2*, and see below). Quantification of chromosomes I, II, and III volumes in hypodermal nuclei revealed an ~45% increase in chromosome volume when animals were exposed to *mes-3* RNAi, suggesting that MES-3 is important for maintaining chromosome territories of hypodermal cells during the normal course of growth and development (*Figure 5d*). Quantification of the degree to which DNA FISH signals for chromosomes I, II, and III overlapped revealed an ~2 fold increase in overlap after *mes-3* RNAi, suggesting that chromosome territories are not just growing, they are also mixing (*Figure 5—figure supplement 1b*). In summary, the data suggest that MES-3 and, therefore, PRC2 is needed to establish and/or maintain chromosome territories in *C. elegans*. Additional studies will be needed to understand the source of cell type specificity of *mes-3* knockdown on genome architecture and related studies using mutant alleles of *mes-3*, as well as loss-of-function alleles in other components of the PRC2, will be needed to confirm the link between PRC2 and the maintenance of chromosome territories.

## Discussion

The invariant cell lineage, transparency, and small genome (100 Mb) of *C. elegans* make this animal an excellent system in which to explore how genome architecture relates to gene expression, development, growth, reproduction, and aging. DNA FISH experiments in *C. elegans* have historically relied on 1) labeling PCR products that cover a single small (5–10 kb) region, or 2) YACs to generate probes targeting larger regions (up to whole chromosomes). Such approaches are low throughput and rigid in the sense that new probe sets need to be produced for each new DNA FISH experiment. Such experiments have also been limited by the types of cells that can be queried, as most DNA FISH protocols rely on dissection of tissues, which is low throughput and limits the number of cell types that can be analyzed at one time. Here we describe an Oligopaint DNA FISH library and

hybridization strategy that allow for visualization of all six *C. elegans* chromosomes at varying scales. The ability to rapidly and cheaply produce *C. elegans* DNA FISH probes, in conjunction with improvements to hybridization protocols, enables DNA FISH in all/most cells across all stages of development in an intact animal. These improvements should empower studies asking if/how higher-order chromatin structures regulate, and/or are regulated by, changes in gene expression that occur during growth and development. Given the invariant cell lineage of *C. elegans*, it should now also be possible to ask if chromosome- chromosome interactions or homolog pairing, or the size, morphology, or sub-nuclear positioning, of chromosomal territories (or subregions of these territories) vary predictably by cell type, age, or developmental trajectory.

# Materials and methods

**Key resources table**

| Reagent type (species) or resource | Designation | Source or reference | Identifiers | Additional information |
|---|---|---|---|---|
| sStrain, strain background (*C. elegans*) | N2 (wild type) | CAENORHABDITIS GENETICS CENTER (CGC) | N/A | https://cgc.umn.edu/strain/N2 |
| sStrain, strain background (*C. elegans*) | CB1489 (him-8 (e1489)) | CAENORHABDITIS GENETICS CENTER (CGC) | WB Cat# CB1489, RRID:WB-STRAIN:CB1489 | https://cgc.umn.edu/strain/CB1489 |
| Strain, strain background (*C. elegans*) | CB1370 (daf-2 (e1370)) | CAENORHABDITIS GENETICS CENTER (CGC) | WB Cat# CB1370, RRID:WB-STRAIN:CB1370 | https://cgc.umn.edu/strain/CB1370 |
| Strain, strain background (*E. coli*) | OP50 | CAENORHABDITIS GENETICS CENTER (CGC) | WB Cat# OP50, RRID:WB-STRAIN:OP50 | https://cgc.umn.edu/strain/OP50 |
| Genetic reagent (*E. coli*) | HT115 (DE3), control bacteria for RNAi | CAENORHABDITIS GENETICS CENTER (CGC) | WB Cat# HT115(DE3), RRID:WB-STRAIN:HT115(DE3) | https://cgc.umn.edu/strain/HT115(DE3) |
| Genetic reagent (*E. coli*) | mes-3 RNAi clone | PMID: 12828945 | N/A | Ahringer RNAi library |
| Software, algorithm | ImageJ | https://imagej.nih.gov/ij/download.html | ImageJ, RRID:SCR_003070 | See *Supplementary file 8* |
| Software, algorithm | 3D Objects Counter Plugin Imagej | https://imagej.net/3D_Objects_Counter | 3D Objects Counter, RRID:SCR_017066 | See *Supplementary file 8* |
| Software, algorithm | 3D ROI Manager | http://imagejdocu.tudor.lu/doku.php?id=plugin:stacks:3d_roi_manager:start | N/A | See *Supplementary file 8* |

## Design and synthesis of multiplexed DNA FISH library

We used a previously described pipeline to mine the *C. elegans* genome (build *ce10*) for desirable (see main text) oligonucleotide nucleotide sequences 42 base pairs in length (*Supplementary file 1*) (*Beliveau et al., 2012*). 872,946 oligonucleotide sequences met this criteria, and 170,594 probes sequences were chosen for the library (*Supplementary file 2*). A series of barcode sequences were appended to each 42 bp hybridization sequence, which resulted in each primary probe being 150 bp. Barcode sequences can be found in *Supplementary file 5*. The 170,594 sequences were ordered as two 90 k oligonucleotide chips from Custom Array (Bothell, WA). Note that a step by step protocol for Oliogpaint probe synthesis can be found in *Supplementary file 6*. To obtain primary probes for Oligopaint experiments, desired primary probes were first amplified using primers specific to the outermost barcode sequences, which correspond to the individual chromosome barcodes shown in *Figure 1a* (*Supplementary file 5*). Single stranded probe (primary probe) generation

was conducted as previously described (*Chen et al., 2015*). Briefly, PCR was used to append a T7 polymerase site to the 5' end of chromosome specific barcode sequence, followed by T7 polymerase reactions to generate ssRNA. ssRNA was reverse transcribed into ssDNA. Unwanted ssRNA species were degraded using base hydrolysis. Finally, long ssDNA oligos were purified using the Zymo-100 DNA Clean and Concentrator Kit with oligo binding buffer. Probes were stored at 100 pmol/ul at −20C. An aliquot of the library is available to qualified labs upon request.

## Bridge and detection probes
Bridge oligos were ordered from IDT at 25 or 100 nmole scales using standard desalting procedures. Fluorescent detection oligos were ordered from IDT with 5' and 3' fluorescent modifications on the 250 nm or 1 um scale and subjected to HPLC purification. Bridge and detection probe sequences are listed in *Supplementary file 5*.

## DNA FISH
See *Supplementary file 7* for detailed protocol on worm collection and Oligopaint FISH. 10 cm plates containing adult (or mixed stage) *C. elegans* were washed with M9 solution (11 mM $KH_2PO_4$, 21 mM $Na_2HPO_4$, 4 mM NaCl, 9 mM $NH_4Cl$ in $H2O$) and collected in 15 ml conical tubes. Animals were pelleted (3 k rpm for 30 s), and washed two times with M9 solution. Animals were resuspended in 10 ml of M9 solution and rocked for ~30 min at room temperature. Animals were pelleted and aliquoted to 1.5 ml microcentrifuge tubes (30–50 ul of packed worms per tube). Samples were placed in liquid nitrogen for 1 min. Frozen worm pellets were resuspended in cold 95% ethanol and vortexed for 30 s. Samples were rocked for 10 min at room temperature. Samples were spun down (3 k rpm for 30 s), supernatant discarded, and washed twice in 1X PBST (10X Phosphate- Buffered Saline (Thermo Fisher Scientific: 70011–044) diluted to 1X in H2O, 0.5% Triton X-100 (Sigma: X100). 1 ml of 4% paraformaldehyde solution (4% paraformaldehyde in 1X PBS) was added and samples were rocked at room temperature for 5 min, washed twice with 1X PBST, and resuspended in 2X SSC (20X saline-sodium citrate (SSC) buffer (Thermo Fischer Scientific: 15557–044) diluted in H2O) for 5 min at room temperature. Samples were spun down and resuspended in a 50% formamide 2X SSC solution at room temperature for 5 min, 95°C for 3 min, and 60°C for 20 min. Samples were spun and resuspended in 60 ul of hybridization mixture (10% dextran sulfate, 2X SSC, 50% formamide, 100 pmol of primary probe per chromosome and 2 ul of RNAse A (sigma 20 mg/ml)). Hybridization reactions were incubated in a 100°C heat block for 5 min before overnight incubation at 37°C in a hybridization oven. The next day, samples were washed with prewarmed 2X SSCT (2X SSC with 0.5% Triton X-100) (rotating at 60°C) for 5 min, followed by a second 2X SSCT wash at 60°C for 20 min. Wash buffer was removed and samples were resuspended in 60 ul of bridge oligo hybridization mixture (2X SSC, 30% formamide, 100 pmol of bridge oligo per targeted region (ie whole chromosome, three megabase, or 500 kb spots) and 100 pmol of each detection oligo. Bridge/detection oligo hybridization reactions were incubated at room temperature for 3 hr. Samples were washed in prewarmed 2XSSC at 60°C for 20 min, followed by a 5-min wash with 2XSSCT at 60°C and a 20 min wash in 2XSSCT at 60°C. Samples were then washed at room temperature in 2XSSCT. Wash buffer was removed and samples were resuspended in mounting medium (Vectashield with DAPI or Slowfade Gold with DAPI). Samples were mounted on microscope slides and sealed with nail polish.

## Alternate embryo DNA FISH protocol
DNA FISH on *in utero* embryos was only 50% efficient. The following protocol improves this efficiency to 90%. This protocol is an adaptation of an existing *C. elegans* DNA FISH protocol (*Crane et al., 2015*). Briefly, adults were dissected in 8 ul of 1X egg buffer on a coverslip (25 mM HEPEs, pH 7.3, 118 mM NaCl2, 48 mM KCl, 2 mM CaCl2, 2 mM MgCl2) to release embryos. Coverslips were placed on a Superfrost Plus Gold slide (Thermo Scientific) and placed in liquid nitrogen for 1 min. Coverslips were popped off with a razor blade and slides were submerged in 95% cold (−20C) ethanol for 10 min. Slides were washed twice in 1XPBST before fixation in 4% paraformaldehyde solution (described above) for 5 min. Slides were washed twice in 1XPBST. 20 ul Primary hybridization mixture (described above) was added to each sample and a coverslip was placed on top. Slides were placed on a 90°C heat block for 10 min. Slides were placed in a humid chamber at 37°C overnight. Wash steps and bridge/detection oligo hybridization was carried out as described

above. 15 ul of mounting medium was added to each sample, and coverslips were sealed with nail polish.

## Microscopy

Standard fluorescent microscopy was conducted on a widefield Zeiss Axio Observer.Z1 microscope using a Plan-Apochromat 63X/1.40 Oil DIC M27 objective and an ORCA-Flash 4.0 CMOS Camera. The Zeiss Apotome 2.0 was used for structured illumination microscopy using three phase images. All image processing were done using the Zen imaging software from Zeiss. Confocal microscopy was done using a Nikon Eclipse Ti microscope equipped with a W1 Yokogawa Spinning disk with 50 um pinhole disk and an Andor Zyla 4.2 Plus sCMOS monochrome camera. A 60X/1.4 Plan Apo Oil objective or a 100X/1.45 Plan Apo Oil objective was used.

## Aging assay

Ten adult animals were picked to 6 cm NGM plates seeded with OP50, and 10 plates were used for each condition. Adult animals were picked off 24 hr later and sacrificed. Once the offspring reached the fourth larval stage, 50 animals were transferred to 6 cm NGM plates seeded with OP50 that were soaked in 1 ml of FUDR solution (3 mg of FUDR (abcam) per plate) the previous day. 20 FUDR soaked plates were used per condition. After 24 hr, 10 plates per condition were collected and animals were frozen as pellets in liquid nitrogen before storage at −80C (Day 1 adult samples). Ten days later, the same collection was repeated on the remaining 10 plates per condition (Day 10 adult samples). Dead animals, as determined by animals that did not respond to a light touch, were removed prior to sample collection for each condition.

## RNAi assay

Wild-type (N2) embryos were collected via hypochlorite treatment (see *Supplementary file 7* for description of embryo isolation by hypochlorite treatment) and placed on RNAi plates (NGM plates with 2.5 mM $KH_2PO_4$27 mM Carbenicillin, 1 mM IPTG) seeded with either HT115 bacteria, or HT115 bacteria expressing *mes-3* dsRNA for two generations: the embryos of adult animals grown on either treatment were placed back onto either treatment, grown to adulthood, and collected for FISH analysis (see *Supplementary file 7* for description of sample collection). The *mes-3* RNAi clone was obtained from the Ahringer library and confirmed to target *mes-3* by Sanger sequencing (*Kamath and Ahringer, 2003*).

## Assessment of chromosome territory volumes and overlapping chromosome territory volumes

For a step-by-step protocol for the image analysis used in this study see *Supplementary file 8*. All territory quantifications were done using standard tools in ImageJ along with the 3D objects counter plugin (*Bolte and Cordelières, 2006*). First, each individual nucleus was segmented from the original file to generate individual nuclei files. The four-channel stack was then split to create individual files for each chromosome (each chromosome is represented by a single fluorophore/channel). To remove background noise and create a binary mask, each image was subjected to thresholding using the default ImageJ thresholding using 'auto' across every image. Once masks were obtained the 3D objects counter tool was utilized to select objects larger than 30 voxels (eliminating further background signal). Object masks for each channel were loaded into the 3D Manager plugin for ImageJ, and all objects for a given chromosome were merged into a single object. The colocalization and measure 3D functions within 3D manager were used to determine the volume of each chromosome as well as the volume of overlap between each chromosome.

## Acknowledgements

We thank members of the Kennedy and Wu Labs for helpful discussions. We thank Ting Wu for helpful discussions and manuscript advice. We thank Barbara Meyer and Satoru Uzawa for discussions on DNA FISH in *C. elegans* embryos. We would like to thank Paula Montero Llopis of the MicRoN imaging core at HMS for microscope assistance. BF and SN were supported by NSF graduate research fellowships. This work was supported by the National Institutes of Health: 5DP1GM106412,

R01HD091797, R01GM123289 awarded to C.-ting Wu for GN and SN, and RO1GM088289 awarded to SK for BF and SK.

## Additional information

### Funding

| Funder | Grant reference number | Author |
|--------|------------------------|--------|
| National Institutes of Health | RO1GM088289 | Scott Kennedy |
| National Institutes of Health | 5DP1GM106412 | Son C Nguyen Guy Nir |
| National Institutes of Health | R01HD091797 | Son C Nguyen Guy Nir |
| National Institutes of Health | R01GM123289 | Son C Nguyen Guy Nir |

The funders had no role in study design, data collection and interpretation, or the decision to submit the work for publication.

### Author contributions

Brandon D Fields, Conceptualization, Data curation, Formal analysis, Validation, Investigation, Visualization, Methodology, Writing—original draft, Writing—review and editing; Son C Nguyen, Conceptualization, Formal analysis, Investigation, Methodology, Writing—review and editing; Guy Nir, Data curation, Formal analysis, Methodology, Writing—review and editing; Scott Kennedy, Conceptualization, Funding acquisition, Investigation, Methodology, Writing—original draft, Project administration, Writing—review and editing

### Author ORCIDs

Brandon D Fields (ID) http://orcid.org/0000-0002-8410-9374
Guy Nir (ID) http://orcid.org/0000-0001-9268-6596
Scott Kennedy (ID) http://orcid.org/0000-0002-7974-8155

### Decision letter and Author response

Decision letter https://doi.org/10.7554/eLife.42823.029
Author response https://doi.org/10.7554/eLife.42823.030

## Additional files

### Supplementary files

• Supplementary file 1. All available 42mer oligos mined with the Oligopaint pipeline. Using the Oligopaint pipeline described in *Beliveau et al. (2012)*, suitable 42mer oligos for FISH were designed and screened against the ce10 genome. The chromosome, start coordinate, end coordinate, sequence, and melting temperature are provided.
DOI: https://doi.org/10.7554/eLife.42823.019

• Supplementary file 2. Selected oligos used for *C. elegans* Oligopaint. A subset of oligos were selected to create the Oligopaint library. The chromosome, start coordinate, end coordinate, sequence, and melting temperature are provided.
DOI: https://doi.org/10.7554/eLife.42823.020

• Supplementary file 3. Order file for first set of oligos synthesized by CustomArray. The appropriate primer sequences were concatenated to the original 42mer sequences and sent for ordering. The chromosome, start coordinate, and full probe sequence is provided.
DOI: https://doi.org/10.7554/eLife.42823.021

• Supplementary file 4. Order file for second set of oligos synthesized by CustomArray. The appropriate primer sequences were concatenated to the original 42mer sequences and sent for ordering. The chromosome, start coordinate, and full probe sequence is provided.

DOI: https://doi.org/10.7554/eLife.42823.022

• Supplementary file 5. Primer, barcode, bridge, and detection oligo sequences used in this study. This file contains the barcodes, bridge, and detection oligos used in this study, as well as the fluorophores used to label each detection oligo. Similar bridge and detection oligos could in theory be used for any primary probe targeting any genome.

DOI: https://doi.org/10.7554/eLife.42823.023

• Supplementary file 6. Step-by-step Oligopaint probe synthesis protocol.

DOI: https://doi.org/10.7554/eLife.42823.024

• Supplementary file 7. Step-by-step sample collection and *C. elegans* Oligopaint protocol.

DOI: https://doi.org/10.7554/eLife.42823.025

• Supplementary file 8. Step-by-step image analysis workflow for chromosome territory volume and overlap.

DOI: https://doi.org/10.7554/eLife.42823.026

• Transparent reporting form

DOI: https://doi.org/10.7554/eLife.42823.027

### Data availability

All data generated or analysed during this study are included in the manuscript and supporting files.

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
