## [Decision Letter]

Thank you for submitting your article "A Multiplexed DNA FISH strategy for Assessing Genome Architecture in *C. elegans*" for consideration by *eLife*. Your article has been reviewed by four peer reviewers, including Brian D Slaughter as the Reviewing Editor and Reviewer #4, and the evaluation has been overseen by Jessica Tyler as the Senior Editor. The following individuals involved in review of your submission have agreed to reveal their identity: Susan Strome (Reviewer #1); Barbara Panning (Reviewer #3).

The reviewers have discussed the reviews with one another and the Reviewing Editor has drafted this decision to help you prepare a revised submission.

Summary:

This manuscript details the adaptation of DNA chromosome Oligopaints for use in *Caenorhabditis elegans*. By altering the hybridization strategy from other DNA oligo-paint protocols, the authors have come up with a bridge oligo method that allows for improved flexibility in the labeling of the FISH oligo pools. Fields et al. designed a set of ~170,000 *C. elegans* oligos arrayed on two 90,000-oligo chips. Each has a unique 42 bp primary oligo plus 3 barcodes, 1 for that oligo's chromosome, 1 for that oligo's 3 MB location, and 1 for that oligo's 0.5 MB location. Using the library of composite oligos in combination with bridge oligos and detection oligos conjugated to 1 of 3 fluorophores, the investigators can perform DNA FISH with up to 6 combinations to image whole chromosomes and/or 0.5-3 MB regions of chromosomes.

Having established the effectiveness and versatility of the method, the authors then use this method to assess chromosome organization during aging. They show that chromosomal territories become less distinct (enlarged and disorganized) with more overlap (suggesting that chromosomal territories are not maintained) and that this disorganization is age and not time related. Finally, they identify one molecular player, *mes-3*, knockdown of which phenocopies age-related chromosome disorganization.

This manuscript will be of broad interest – the FISH method will be useful for the *C. elegans* community and the finding that chromosome organization is altered with age will be intriguing to the aging community. Overall, the manuscript is a nice methods paper that will provide a useful resource for broad use within the *C. elegans* community to address key questions about how genome architecture influences multiple processes and cell types within a developing and aging organism. The reviewers shared enthusiasm for the paper, and would be supportive of publication given that the concerns below are addressed.

Essential revisions:

In Figure 2, more work should be given to demonstrate the specificity of the Oligopaint approach. To demonstrate specificity, Oligopaint could be done on animals in which oocytes carry a fusion of two chromosomes – painting one of the chromosomes of the fusion should paint only half of the long fused chromosome. Probably easier would be to paint the X chromosomes in *him-8* mutant oocytes, in which the X chromosomes exist as univalents instead of a bivalent.

STORM data:

Subsection “Using OligoSTORM to super resolve *C. elegans* chromosomes within whole animals”: The authors need to be cautious using FISH and STORM to quantify particle density. First, there is no way to know if all the probes are getting activated since STORM is a stochastic method. Thus, an unknown percent of the probes could remain in the dark state and never be activated; or probes might not be deactivated and sent back to the dark state. Second, there is no way to know if all of the probes have hybridized, thus it is dangerous to assume that the probes are still equally distributed once they have been hybridized in the worm. The authors should address both of these in the manuscript.

Subsection “*C. elegans* Oligopaint library design: The authors mention that the oligos are "evenly spaced" with about 2 probes per kilobase of genomic DNA. Is this true all the way across the chromosome? In other organisms the probes are on average evenly spaced, but there are some regions where this can't occur because of repetitive DNA or potential crosstalk of the sequence with another chromosome. Furthermore, the authors use this equal distribution of probes to quantify the density of DNA in the STORM experiment. If regions of the genome are not evenly covered, then that needs to be mentioned and addressed in the STORM quantification.

Beyond this point, the authors should make more clear what the utility of the STORM analysis is, and what specially it adds to the manuscript. The chiasmata between homologs is easily represented with standard fluorescent microscopy. If true, the 'core' idea would be of interest. However, STORM is very sensitive to environment, and more work should be done to determine that the 'core' at the center of each chromosome IV homolog is real, and not an artifact of STORM. It needs to be verified with a further method. The size of the gradient of STORM density is on the order of a micron. It true, standard, high resolution, confocal data should be able to see this same gradient. Using one color to mark the chromosome, followed by DAPI quantification, may also see it. Is there evidence of such a 'core' density difference from Hi-C data?

If the authors can demonstrate that this core is missing from other chromosomes, or is missing in some condition such as aged *C. elegans*, it would also go a long way in determining that this is not an artifact of STORM.

Are the authors able to use STORM to resolve any detail on the scale (reported as 80 nm) of their improved resolution? And if so, what is the relationship between what conclusions can be made with super resolution imaging yet with relatively low probe density (2 probes per kb)? Further discussion on this point would be useful.

More thorough explanation of methods and more discussion:

Some sections of Materials and methods are quite minimal. Especially since this is a Tools and Resources article, the Materials and methods should provide protocols that are detailed enough for others to use the technology.

Specifically, the authors combined 3 colors in different combinations to come up with 6 signatures for the 6 chromosomes (Figure 3A). This is very effective in outlining the 6 chromosomes in cases where there is either separation of the chromosomes, or very discrete chromosome territories. However, in cases where there are not discrete territories, and overlap between chromosomes becomes dramatic, separation of 6 chromosomes with 3 colors becomes very difficult. The authors acknowledge this in the Figure 6 figure legend – and quantification in Figure 6B and CD was done on 3 chromosomes only because of this difficulty. This should be explained more thoroughly in the text.

The separation of chromosomes with this method, especially in circumstances of chromosome overlap, is a key, others will want to do this in mutants, aged cells etc. Part of this being considered as a tool and resource is a full description of methods not just for probe design, but data analysis. More explanation is needed on criteria for considering a region positive or negative for a color, and how the final determination of where a chromosome is, in cases of high degree of overlap, should be discussed with rigor so others can reproduce it.

In the section with the *mes-3* RNAi – while the images in aging intestinal nuclei and *mes-3* hypodermal nuclei look relatively similar, the quantitation of overlap was quite different. Is that a function of different cell types? Or does it reflect that different mechanisms underlie the breakdown of nuclear organization.

---

## [Author Response]

Essential revisions:In Figure 2, more work should be given to demonstrate the specificity of the Oligopaint approach. To demonstrate specificity, Oligopaint could be done on animals in which oocytes carry a fusion of two chromosomes – painting one of the chromosomes of the fusion should paint only half of the long fused chromosome. Probably easier would be to paint the X chromosomes in him-8 mutant oocytes, in which the X chromosomes exist as univalents instead of a bivalent.

We conducted the suggested experiment and the data are included in our revised manuscript as Figure 2—figure supplement 1 and are pasted below. The data confirm the specificity of the library. Text has been modified accordingly:

“HIM-8 is required for X chromosome homolog pairing during meiosis and, consequently, X chromosomes are present as two univalents (and not a single bivalent) in *him-8* mutant animals (Phillips et al., 2005). […] As expected, X chromosome DNA FISH stained a single bivalent in wild-type oocytes (58/58) and two univalents in 84% of *him-8* oocytes (51/61) (Figure 2—figure supplement 1).”

STORM data:Subsection “Using OligoSTORM to super resolve C. elegans chromosomes within whole animals”: The authors need to be cautious using FISH and STORM to quantify particle density. First, there is no way to know if all the probes are getting activated since STORM is a stochastic method. Thus, an unknown percent of the probes could remain in the dark state and never be activated; or probes might not be deactivated and sent back to the dark state. Second, there is no way to know if all of the probes have hybridized, thus it is dangerous to assume that the probes are still equally distributed once they have been hybridized in the worm. The authors should address both of these in the manuscript.Subsection “C. elegans Oligopaint library design: The authors mention that the oligos are "evenly spaced" with about 2 probes per kilobase of genomic DNA. Is this true all the way across the chromosome? In other organisms the probes are on average evenly spaced, but there are some regions where this can't occur because of repetitive DNA or potential crosstalk of the sequence with another chromosome. Furthermore, the authors use this equal distribution of probes to quantify the density of DNA in the STORM experiment. If regions of the genome are not evenly covered, then that needs to be mentioned and addressed in the STORM quantification.

We thank the reviewers for raising concerns we had not considered. We reanalyzed the distribution of our probe sets on a log2 scale and noted subtle differences in probe density between the center and arms of chromosomes, which could conceivably explain the chromosome “core” phenomena we described in our original submission. Also, we noticed “core-like” structures using other microscopy approaches (see below), however, these structures were also present in non-biological structures. The new data raise concerns that cores are real. Obviously more work is needed before the STORM data can be included in the paper. The amount of this work is substantial and I am not in a position to have the work done at the moment (first author has left the lab for a postdoc position and I don’t have anyone in the lab to replace him at the moment). Because STORM was not even mentioned by our reviewers in the synopsis of our paper, or listed as a strength of the paper, I’ve decided the best way forward here is simply to remove the data from the revised manuscript. Should editor/reviewers disagree, and prefer that we include the data, we would be happy to do so. In this case, we would put the data in the supplement along with a detailed description of the reasons for and against believing “cores” might be real.

Subsection “C. elegans Oligopaint library design: The authors mention that the oligos are "evenly spaced" with about 2 probes per kilobase of genomic DNA. Is this true all the way across the chromosome?

We have modified text (and showed our data in log2) to better describe the distribution of probes in our revised manuscript. Text was modified as follows:

“The results of our bioinformatic search revealed approximately 9 suitable probe sequences per kilobase of *C. elegans* genomic DNA (Table 1). […] Similar probe distributions are observed for the other five *C. elegans* chromosomes (Figure 1—figure supplement 1).”

Beyond this point, the authors should make more clear what the utility of the STORM analysis is, and what specially it adds to the manuscript. The chiasmata between homologs is easily represented with standard fluorescent microscopy. If true, the 'core' idea would be of interest. However, STORM is very sensitive to environment, and more work should be done to determine that the 'core' at the center of each chromosome IV homolog is real, and not an artifact of STORM. It needs to be verified with a further method. The size of the gradient of STORM density is on the order of a micron. It true, standard, high resolution, confocal data should be able to see this same gradient. Using one color to mark the chromosome, followed by DAPI quantification, may also see it. Is there evidence of such a 'core' density difference from Hi-C data?If the authors can demonstrate that this core is missing from other chromosomes, or is missing in some condition such as aged C. elegans, it would also go a long way in determining that this is not an artifact of STORM.Are the authors able to use STORM to resolve any detail on the scale (reported as 80 nm) of their improved resolution? And if so, what is the relationship between what conclusions can be made with super resolution imaging yet with relatively low probe density (2 probes per kb)? Further discussion on this point would be useful.

We did do the suggested DAPI and traditional DNA FISH experiments and we observed core-like structures in *C. elegans* chromosomes. During the course of these experiments, however, we also noticed “core-like” structures (using these same approaches) in fluorescent beads. We can’t think of any reason why fluorescent beads should have cores. For this reason (and because of the subtly non-random distribution of probe sets shown in new Figure 1A) we’ve decided it is best to tap the breaks on chromosome “cores” for the time being. Therefore, STORM is no longer a part of the revised manuscript. We thank our reviewers for pointing us in the right direction here.

More thorough explanation of methods and more discussion:Some sections of Materials and methods are quite minimal. Especially since this is a Tools and Resources article, the Materials and methods should provide protocols that are detailed enough for others to use the technology.

We added some information to our Materials and methods. The main route we have taken, however, to address this concern was by adding three new supplementary files (Supplementary files 6, 7, 8), which contain details for probe synthesis (Supplementary file 6), DNA FISH (Supplementary file 7), and a step by step image analysis workflow including screenshots of every major step in the process (Supplementary file 8). The supplementary files also describe necessary equipment, reagents (including catalog numbers and buffer compositions), and software required to carry out the experiments described in this manuscript. We believe that this format should allow for an easier adaption of the technology described here for other labs.

Specifically, the authors combined 3 colors in different combinations to come up with 6 signatures for the 6 chromosomes (Figure 3A). This is very effective in outlining the 6 chromosomes in cases where there is either separation of the chromosomes, or very discrete chromosome territories. However, in cases where there are not discrete territories, and overlap between chromosomes becomes dramatic, separation of 6 chromosomes with 3 colors becomes very difficult. The authors acknowledge this in the Figure 6 figure legend – and quantification in Figure 6B and CD was done on 3 chromosomes only because of this difficulty. This should be explained more thoroughly in the text.The separation of chromosomes with this method, especially in circumstances of chromosome overlap, is a key, others will want to do this in mutants, aged cells etc. Part of this being considered as a tool and resource is a full description of methods not just for probe design, but data analysis. More explanation is needed on criteria for considering a region positive or negative for a color, and how the final determination of where a chromosome is, in cases of high degree of overlap, should be discussed with rigor so others can reproduce it.

We have modified the revised manuscript in three ways to address this issue.

First, we have modified the main text to do a better job explaining how we can visualize 6 colors with three fluorophores.

*“C. elegans* possess six chromosomes. […] Overlapping Alexa 647 (green) or Cy3 (red) channels produces a yellow pseudocolor that can be differentiated from Alexa 647 (green) or Cy3 (red) alone (Figure 3—figure supplement 1B).”

Second, we have added Figure 3—figure supplement 1 to clearly illustrate the principle of multicolor labeling.

Third, we have added a few sentences to the end of our 6 chromosome FISH section to explain why defining the space occupied by 6 chromosomes simultaneously using 3 colors is not possible. Finally, our new Supplementary files (6-8) contain information needed to reproduce the types of DNA FISH analyses that are described in our paper.

“Note, because the 6 color strategy described above visualizes overlap between two colors (each of which is also used to define a separate chromosome), rigorously defining the subnuclear space occupied by individual chromosome is not possible using the 6 color approach. Single channel probe sets are recommended for experiments in which knowing the precise space occupied by a chromosome is relevant (see Figure 5 for examples).”

In the section with the mes-3 RNAi – while the images in aging intestinal nuclei and mes-3 hypodermal nuclei look relatively similar, the quantitation of overlap was quite different. Is that a function of different cell types? Or does it reflect that different mechanisms underlie the breakdown of nuclear organization.

We reanalyzed our data and found that the nuclear volume (using DAPI) of intestinal cell nuclei is 5-10x that of skin cell nuclei. Thus, there is more total space in intestinal cells for chromosomes to overlap. In other words, it is possible that degree of overlap is the same in both cell types but that the volume of this overlap is quite different, given the different volumes of these nuclei. We think this likely explains why the quantifications of chromosome overlap, which we presented in the original submission, looked different between the two cells types, and we agree that presenting the data as we did was confusing. To better illustrate our main point, we decided to present data showing changes in chromosome sizes after aging or *mes-3* RNAi (instead of degree of overlap) in the main text of the revised manuscript (Figure 5B and D). This analysis shows that there is a similar overall degree of chromosomal expansion in the two cells types (as hinted at by the images), while also showing the differences in chromosome size between the two cell types. The old analysis, which showed that aging or *mes-3* RNAi increased the degree to which chromosomes overlapped (old Figure 6B/D), is now Figure 5—figure supplement 1.